# Effect of Bimetallic Dimer-Embedded TiO_2_(101) Surface on CO_2_ Reduction: The First-Principles Calculation

**DOI:** 10.3390/ma15072538

**Published:** 2022-03-30

**Authors:** Chongyang Li, Cui Shang, Bin Zhao, Gang Zhang, Liangliang Liu, Wentao Yang, Zhiquan Chen

**Affiliations:** 1College of Electric Power, North China University of Water Resources and Electric Power, Zhengzhou 450045, China; lichongyang@ncwu.edu.cn (C.L.); zg15937100954@163.com (G.Z.); 2Hubei Nuclear Solid Physics Key Laboratory, Department of Physics, Wuhan University, Wuhan 430072, China; 3Henan Key Laboratory of Magnetoelectronic Information Functional Materials, School of Physics and Electronics Engineering, Zhengzhou University of Light Industry, Zhengzhou 450002, China; sc0906@zzuli.edu.cn; 4School of Science, Zhongyuan University of Technology, Zhengzhou 450007, China; zhouko2008@163.com; 5Key Laboratory for Special Functional Materials of Ministry of Education, Henan University, Kaifeng 475004, China

**Keywords:** dimer, TiO_2_(101), CO_2_ reduction, metal-embedded, catalyst, first principles

## Abstract

The first-principles calculation was used to explore the effect of a bimetallic dimer-embedded anatase TiO_2_(101) surface on CO_2_ reduction behaviors. For the dimer-embedded anatase TiO_2_(101) surface, Zn-Cu, Zn-Pt, and Zn-Pd dimer interstitials could stably stay on the TiO_2_(101) surface with a binding energy of about −2.36 eV, as well as the electronic states’ results. Meanwhile, the results of adsorption energy, structure parameters, and electronic states indicated that CO_2_ was first physically and then chemically adsorbed much more stably on these three kinds of dimer-embedded TiO_2_(101) substrate with a small barrier energy of 0.03 eV, 0.23 eV, and 0.12 eV. Regarding the reduction process, the highest-energy barriers of the CO_2_ molecule on the Zn-Cu dimer-embedded TiO_2_(101) substrate was 0.31 eV, which largely benefited the CO_2_-reduction reaction (CO_2_RR) activity and was much lower than that of the other two kinds of Zn-Pt and Cu-Pt dimer-TiO_2_ systems. Simultaneously, the products CO* and *O* of CO_2_ reduction were firmly adsorbed on the dimer-embedded TiO_2_(101) surface. Our results indicated that a non-noble Zn-Cu dimer might be a more suitable and economical choice, which might theoretically promote the designation of high CO_2_RR performance on TiO_2_ catalysts.

## 1. Introduction

Global warming is a serious challenge for human activity today; it is also a global environmental issue recognized by the international community [1,2]. The main reason for global warming is the results of the rise in global population [3,4,5], solid wastes [6], as well as the correspondingly dramatic and continuing uptrend in the concentration of greenhouse gases worldwide, of which carbon dioxide (CO_2_) is the largest contributor [7]. Three kinds of CO_2_-conversion strategies exist: electrocatalysis, thermal-catalysis, and photocatalysis, to the direct conversion of CO_2_ into valuable fuel and chemical products under ambient conditions [8,9,10,11], which has attracted increasing attention from researchers [12,13,14,15,16]. However, the CO_2_-reduction reaction (CO_2_RR) still suffers from low faradaic efficiency, low conversion efficiency, sluggish kinetics of the primary side reaction of hydrogen-evolution reaction, and high overpotential, which have greatly hindered its further practical applications. Based on this, transition-metal (TM)-based catalysts have been applied to improve the CO_2_RR performance [17], as well as waste-originated biorenewables and other catalysts [18,19,20,21,22].

The well-noted single-atom catalyst (SAC) was first attempted to design and synthesize novel materials by defect, modification, high-temperature shockwave, and self-assembly, which have enabled various catalytic conversions, including CO_2_RR with extremely higher atom efficiency, activity, and durability [23,24,25,26,27]. Double-atom catalysts (DACs) such as Cu-Pt [28,29,30,31], Cu-Au [32,33,34], and Pd-Au [35] in many mainstream reactions have also been experimented, characterized, and developed to improve CO_2_-reduction-reaction efficiency, offering a fascinating catalyst model for theorists to explore, and hence gaining enormous attention. DACs usually exhibit better catalytic performance than SACs [28,34,35]. This is mainly due to their tuning diversity and synergistic effect between adjacent active sites of metal dimers, resulting in low-coordination metal atoms and an extra metal site. The interaction of different metal sites could accelerate the intrinsic CO_2_RR performance of bimetallic dimer-embedded catalysts. For example, Baldoví et al. [34] loaded Au-, Cu-, and Au/Cu-alloy NPs on the surface of TiO_2_. The study showed that the introduction of Cu into Au/Cu-TiO_2_ could boost the transfer of the CB electrons in TiO_2_ to CO_2_, resulting in a higher conversion efficiency toward CO_2_ reduction to CH_4_. Zheng et al. [35] introduced a secondary metal (Au) to support the Pd catalyst, and Au concentration could alter the dominant reaction pathway, which was much more favorable for CO_2_ reduction performance. Therefore, the combination of non-noble metals and noble metals and even the non-noble metal dimers embedded on TiO_2_ substrate might be a novel and effective method to promote CO_2_ adsorption and conversion.

Meanwhile, many studies [36,37,38,39] found that CO was an important intermediate in the transformation of CH_4_, CH_3_OH, and HCHO. The CO_2_-reduction reaction can be regarded as a “series reaction”. CO_2_ molecules need to be converted into CO first to obtain the target gas product. Therefore, it is a feasible strategy to design bimetallic dimer TiO_2_ catalysts that can firmly adsorb CO to achieve CO_2_ reduction to CH_4_.

Based on the first-principles calculation, we explored CO_2_ reduction on TiO_2_ with M-N (M, N = Cu, Pt, Zn) dimer doping. We mainly calculated both the binding energy and charge distributions of the bimetallic dimer-embedded A-TiO_2_(101) substrate and the CO_2_ molecule adsorbed on these dimer-embedded A-TiO_2_(101) catalysts, respectively. We also calculated the Bader charge and Charge Density Difference (CDD) from the TiO_2_ substrate to the bimetallic dimer and from the bimetallic dimer-embedded TiO_2_ to CO_2_ gas molecule. The results indicated that the structure of the bimetallic dimer-embedded TiO_2_(101) surface was stable, and the CO_2_ molecule could stably stay on the bimetallic dimer-embedded TiO_2_(101) surface. Then, the CO_2_ reduction reactivity was proposed. The energy barrier of the Zn-Cu dimer-embedded TiO_2_(101) surface was as low as 0.31 eV, which was a much more suitable choice as well as advantageous in terms of economy. Therefore, the findings of this study might theoretically provide an effective strategy for the designing of effective CO_2_-reduction TiO_2_-based catalysts.

## 2. Computational Details

Based on the density functional theory, all first-principles calculations were used in the CO_2_ adsorption and reduction on the bimetallic dimer-embedded A-TiO_2_(101) surface. Projected augmented-wave pseudopotentials, generalized-gradient-approximation (GGA) method [40,41,42], and Perdew–Burke–Ernzerhof (PBE) functional were employed using Vienna Ab-initio Simulation Package (VASP, vasp.5.4.1, Hafner team of University of Vienna, Vienna, Austria) [43,44,45] through the computational simulations. For the kinetic processes, the previous studies indicated that the GGA-PBE approach could correctly describe CO-oxidation, CO_2_-dissociation, and O-diffusion [46,47] reaction paths on the surface. The cutoff of the kinetic energy for the plane wave was set to be 450 eV. Further, 2 × 2 × 1 was used for the Monkhorst–Pack k-point mesh during geometry optimizations, and the corresponding electronic structure calculations were set as 3 × 3 × 1. The convergence criterion for the electronic was chosen as 10^−5^ eV, and the maximum force of each atom for ionic relaxation was 0.02 eV/Å. Regarding the dissociation and diffusion of the CO_2_ reduction process, the transition state (TS) search was performed by the climbing-image nudged-elastic-band method [48,49,50]. Bader charge analysis was used to obtain the electronic transformation, and charge-density difference was calculated to observe its charge distribution [51].

The lattice parameters of the A-TiO_2_ system were a = 3.830 Å and c = 9.613 Å. A 3 × 2 surface supercell of TiO_2_ with three stoichiometric (TiO_2_)-structure layers was selected, of which the bottom TiO_2_ layer was fixed to simulate the bulk structure and the total atom was 108, to investigate the CO_2_ adsorption and dissociation performance. In the direction perpendicular to the (101) plane, the thickness of the vacuum layer was set to be 10 Å to eliminate the influence between the adjacent layers.

When Zn-Cu, Zn-Pt, and Cu-Pt dimers were embedded onto the surface of TiO_2_(101), the binding energy ($E_{b}$) of this bimetallic dimer-embedded A-TiO_2_(101) substrate was defined as
(1)$$E_{b}=E_{\mathrm{surface}+\mathrm{dimer}}-E_{\mathrm{surface}}-E_{{\mathrm{TiO}}_{2}}$$
where $E_{\mathrm{surface}+\mathrm{dimer}}$ is the system energy of the bimetallic dimer-embedded TiO_2_; $E_{\mathrm{surface}}$ is the system energy without bimetallic dimer; and $E_{\mathrm{dimer}}$ is the energy of independent bimetallic dimer.

To explore the adsorption performance of CO_2_ molecules on a bimetallic dimer-embedded A-TiO_2_(101) surface, the adsorption energy ($E_{{\mathrm{CO}}_{2}\mathrm{ad}}$) was defined as
(2)$$E_{{\mathrm{CO}}_{2}\mathrm{ad}}=E_{\mathrm{surf}+{\mathrm{CO}}_{2}}-E_{\mathrm{surf}}-E_{{\mathrm{CO}}_{2}}$$
where $E_{\mathrm{surf}+{\mathrm{CO}}_{2}}$ is the system energy of adsorbed CO_2_ molecules on a bimetallic dimer-embedded A-TiO_2_(101) surface; $E_{\mathrm{surf}}$ is the system energy without the adsorbed CO_2_ molecule; $E_{{\mathrm{CO}}_{2}}$ is the energy of the free gas molecules.

## 3. Results and Discussions

### 3.1. Bimetallic Sites on Anatase TiO_2_(101) Surface

We chose three bimetallic combinations, Zn-Cu, Zn-Pt, and Cu-Pt, to observe the CO_2_ adsorption and reduction on a bimetallic dimer-embedded TiO_2_(101) surface. If a bimetallic dimer was embedded on the A-TiO_2_(101) surface, several possible binding sites could be selected. We chose the most suitable embedded sites through serval configurations of metal-embedded A-TiO_2_(101), which are shown in Figure 1. The binding energy of Zn-Cu, Zn-Pt, and Cu-Pt on the surface of A-TiO_2_(101) was −2.34 eV, −3.6 eV, and −2.9 eV, respectively, indicating that bimetallic-dimer interstitials could stably stay on the A-TiO_2_(101) surface, along with bond length of M-N, M-O_2c_, N-Ti and the angle of O_2c_-M-O_2c_, as listed in Table 1.

Bader charge analysis and CDD were also performed to further study the stability of the bimetallic dimer-embedded TiO_2_ surface. As listed in Table 1, about 0.85 e, 0.52 e, and 0.60 e were transferred from the embedded Zn-Cu, Zn-Pt, and Cu-Pt dimers to the A-TiO_2_(101) substrate for these three different configurations (Figure 1), respectively, which were mainly accepted by the neighboring O atoms. The interaction between embedded bimetallic dimers and A-TiO_2_(101) surface led to the charge redistribution and the accumulation of electrons around the Zn-O, Zn-O, Cu-Pt bonds for Zn-Cu, Zn-Pt, and Cu-Pt dimer embedded TiO_2_ systems, respectively, which could be confirmed by CDD, as shown in Figure 1d–f. Further, the partial density of states (PDOS) of three dimer-TiO_2_(101) systems, as shown in Figure 2, revealed many impurity peaks induced by 2p orbital electrons of O atom and 3d orbital electrons of Zn and Cu atoms, indicating that the chemical activity of embedded TiO_2_ surface was relatively stronger. Therefore, the results of PDOS, Bader charge analysis, and CDD demonstrated the formation of strong chemical bonds between Zn-Cu, Zn-Pt, Cu-Pt dimers and neighboring oxygen atoms. Therefore, the A-TiO_2_(101) surface embedded with Zn-Cu, Zn-Pt, and Cu-Pt dimers were chemically stable.

### 3.2. One CO_2_ Molecule Adsorbed on the Bimetal-TiO_2_(101) Surface

When a CO_2_ molecule came close to these three configuration surfaces, it will be adsorbed on the surface. Before further exploring the CO_2_ adsorption and reduction on the bimetal-TiO_2_(101) surface, several adsorbed sites were determined, and the most stable surface models for Zn-Cu, Zn-Pt, and Cu-Pt dimers were obtained, as shown in Figure 3. For these three configurations, the adsorption energies for CO_2_ molecules of −0.15 eV, −0.17 eV, and −0.13 eV, respectively. The calculation results demonstrated weaker physical adsorption between the CO_2_ molecule and bimetallic interstitial. The structure of CO_2_ molecule is nearly the same with the gas state with the bond length of 1.18 Å and ∠O_(1)_-C-O_(2)_ around 180°. At the same time, the results of the charge transfer from A-TiO_2_ to the bimetallic dimer showed few electrons transferring from the CO_2_ molecules to the dimer-embedded TiO_2_ substrate, namely 0.03 e, 0.02 e, and 0.02 e for Zn-Cu, Zn-Pt, and Cu-Pt dimer-embedded TiO_2_(101) systems, respectively. Thus, the interactions between the CO_2_ molecule and the dimer-embedded A-TiO_2_(101) surface were relatively weaker. All these results showed a weaker physical absorption between the system and CO_2_ molecules.

Meanwhile, the CO_2_ gas molecule was more stably adsorbed onto the bimetal-embedded TiO_2_ surface with a small barrier energy, which was about 0.03 eV, 0.23 eV, and 0.12 eV for the Zn-Cu, Zn-Pt, and Cu-Pt series, respectively. The final stable adsorbed structures are shown in Figure 4a–c. Moreover, the charge distribution from CDD is shown in Figure 4d–f. As shown in Figure 4, the CO_2_ molecules were bound to the dimer-TiO_2_(101) system via the bimetallic atoms. The distances between the C atom and the nearest embedded metal of the dimer-TiO_2_ substrate were shortened to be 1.87 Å, 2.00 Å, and 1.99 Å for Zn-Cu, Zn-Pt, and Cu-Pt adsorption cases, respectively, which were much nearer than those of the initial adsorption cases. Moreover, the bonds of O_(1)_-C and C-O_(2)_ were enlarged to be about 1.24 Å and 1.30 Å, respectively, and the adsorption energies were elevated up to −0.36 eV, −0.46 eV, and −0.97 eV for Zn-Cu, Zn-Pt, and Cu-Pt cases, respectively. Therefore, the adsorption of CO_2_ molecules on the dimer-TiO_2_(101) surface should be a chemical adsorption behavior.

The partial densities of states of the absorbed CO_2_ on the bimetallic dimer-embedded TiO_2_ system (bimetallic dimers were Zn-Cu, Zn-Pt, and Cu-Pt) were calculated to further understand this adsorption performance, which are shown in Figure 5. When CO_2_ molecules were adsorbed on the bimetal-embedded TiO_2_ surface, we could clearly observe the hybridization with the orbitals of adsorbed CO_2_ molecules and 3d orbitals of dopant Zn, Cu, and Pt atoms, implying the formation of Zn-O, Cu-O, and Pt-O bonds on the interface between the adsorbed CO_2_ molecules and the dimer-TiO_2_ system. Moreover, the Bader charge analysis revealed about 0.82 e, 0.67 e, and 0.54 e transferred from the CO_2_ molecules to the dimer-embedded TiO_2_ substrate for Zn-Cu, Zn-Pt, and Cu-Pt dimers (see Table 2). In addition to the calculations of charge distribution, we could see that the accumulation of charge occurred around metal-O bonds; the CDD is plotted in Figure 4d–f. The results of the charge transfer and charge redistribution demonstrated the strong chemical interaction between the dopant dimer of Zn-Cu, Zn-Pt, and Cu-Pt and nearby O adatoms, namely Zn-O, Zn-O, and Cu-O bonds. This might be the reason for the increase in O_(1)_-C and C-O_(2)_ bond lengths. Further, we found that the adsorption energy of the Cu-Pt-TiO_2_ system was relatively higher, which was two times more than that of other two cases. The absolute value of charge transfer for the Zn-Cu-TiO_2_ system, as well as the extents of overlap of electronic states was also larger. Therefore, the aforementioned results, especially the structural variation and charge redistribution of CO_2_ molecules and different dimer-TiO_2_ substrates, showed stronger chemical interaction between CO_2_ molecules and dimer-embedded TiO_2_(101) surface. Thus, CO_2_ molecules could stably stay and were chemically adsorbed. The details of initial physical adsorption to final chemical adsorption, along with the reduction process, were as follows.

### 3.3. CO_2_ Reduction on the Zn-Cu, Zn-Pt, and Cu-Pt Dimer-Embedded Anatase TiO_2_(101) Surface

After determining the structure and CO_2_ adsorption of bimetallic dimer-embedded on the A-TiO_2_(101) surface, we first discussed the effect of the Zn-Cu dimer-embedded A-TiO_2_(101) substrate on the reduction of the CO_2_ molecule. We chose the direct reduction pathway. We determined the optimized structures of the TSs across dissociation and diffusion processes, as well as the products involved in this pathway, which are shown in Figure 6. For the Zn-Cu-CO_2_ system, CO_2_ molecules subsequently dissociated into the adsorbed *CO of the O-C-Cu bond and *O* bonded with Cu, Zn, and Ti atoms onto the surface via the transition states TS1, TS2, and TS3 and intermediate states A2 and A3. The asterisk (*) is denoted as the adsorbed sites. When a CO_2_ molecule was adsorbed on the surface of the Zn-Cu dimer-embedded A-TiO_2_(101) surface, the Cu and Zn atoms could diffuse to adsorb the CO_2_ molecule and the Zn-Cu bond was broken, which resulted in the formation of new bonds of Cu-O_(1)_, Cu-C, Cu-Ti, and Zn-O_(2)_. The energy barrier for TS1 was only 0.03 eV, and then the CO_2_ molecule could be stably adsorbed onto the surface of the TiO_2_ system.

As shown in Figure 6b, A2 was the initial configuration and the CO_2_ gas phase dissociated. A lower energy of 0.31 eV was needed for the dissociation step. In this step, the O_(1)_ atom diffused away and led to the breaking of the Cu-O_(1)_ bond, while the O_(2)_ atom moved a little nearer and then bonded with Cu, Zn, and Ti atoms, which were much closer to the A-TiO_2_(101) surface through TS2. With a decrease of 0.84 eV, the Zn-Cu-TiO_2_(101) system achieved a more stable structure than A2 after this transformation. However, A3 was a metastable configuration, since C and O_(2)_ from the gas phase exhibited more polarity. The interaction with the surface Cu atom was enhanced, and the more active atom of embedded Cu easily diffused outward to be adsorbed by OC* to a more stable site by an energetic driving force. With a tiny energy barrier of 0.17 eV, the diffusion of A3 transforming into A4 was achieved easily. As a result, with the breaking of the Cu-Ti bond, the new bonds of O_(1)_-C, C-Cu, and Cu-O_(2)_ were aligned in a straight line with the bond lengths of 1.15 Å, 1.76 Å, and 1.79 Å. For the whole reaction process of A1 → A2 → A3 → A4 (Figure 6), the highest-energy barrier and the total reaction energy of CO_2_ reduction on the Zn-Cu-TiO_2_ system were 0.31 eV and 0.72 eV, respectively, corresponding to the transformation from A2 to A4. The results indicated that the reaction of CO_2_ dissociation became much easier with the help of the Zn-Cu dimer embedded on the A-TiO_2_(101) surface. The embedded dimer might also contribute a more active site on the surface and improve the CO_2_ reduction behaviors. Meanwhile, the configuration structure also became more stable upon CO_2_ adsorption and reduction reaction on the dimer-embedded A-TiO_2_ surface.

After a CO_2_ molecule was adsorbed on the A-TiO_2_(101) surface, the structure B1 was first obtained easily without nearly any difference with the independent CO_2_ gas phase and the Zn-Pt-TiO_2_ system, as shown in Figure 1b, indicating the physical adsorption of the stable structure B1. For the structure B1, the embedded Zn and Pt atoms were particularly prone to adsorb CO_2_ molecules. Then, a much more stable structure B2 was obtained through TS4 with an energy barrier of 0.23 eV (Figure 7a); the system energy slightly decreased by 0.28 eV. As a result, the two new bonds of Pt-C and Zn-O_(2)_ were formed, and the adsorption energy of the CO_2_ molecule increased to −0.46 eV. The adsorbed CO_2_ molecule on the surface became much more stable via chemical adsorption. For B2 as the initial configuration, the adsorbed CO_2_ molecule was dissociated on the Zn-Pt dimer-embedded TiO_2_(101) surface, as shown in Figure 7b. The dissociation energy barrier of the adsorbed CO_2_ molecule was 1.69 eV, which was much higher than that in the Zn-Cu case, as shown in Figure 6b. In the transition state TS5, the adsorbed CO_2_ molecule was dissociated, along with the breaking of C-O_(2)_ and Pt-Ti bonds and the formation of new bonds of Pt-O_(2)_ and Pt-O_3c_ on the subsurface. For the whole reaction path of C1 → TS4 → C2 → TS5 → C3, the highest-energy barrier was 1.69 eV, and the system energy decreased by 0.9 eV. The results also demonstrated that the dissociated CO was stably adsorbed on the Zn-Pt dimer-embedded TiO_2_(101) surface, and the final structure B3 also became much more stable.

For the Cu-Pt case, the structures C1 and C4 were initial and final configurations; the dissociation-reaction energy and energy barrier were −1.34 eV and 1.58 eV, respectively. Similarly to the former two dimer-embedded TiO_2_(101) surfaces, a more stable structure, C2, was obtained with a lower-energy barrier of 0.12 eV. Obviously, the new Pt-C and Cu-O_(2)_ chemical bonds of 1.99 Å and 1.90 Å, were formed. The C-O_(2)_ bond was strengthened to be 1.30 Å with a small bend of CO_2_ molecules, while the Cu-Pt bond was broken. Subsequently, the adsorbed CO_2_ molecule began to dissociate, as shown in Figure 8b. In the transition state TS7, the C-O_(2)_ bond of the CO_2_ molecule was initially broken, with a dissociation-energy barrier of 1.58 eV. As a result, O_(2)_ atoms moved nearer to the TiO_2_(101) surface, and a relatively more stable structure C3 was obtained, along with the formation of new bonds of O_(2)_-Cu, O_(2)_-Pt, and O_(2)_-Ti. At the same time, C3 was a metastable configuration. The Pt atom could easily move out to be grabbed by the *CO adatom during TS7 and TS8 with the distance varying from 1.99 Å, 1.85 Å to 1.82 Å via an energetic driving force. Via the diffusion of TS8, the embedded Pt atom of structure C4 was more attracted to the *CO adatom with exothermic energy and energy barrier of 0.18 eV and 0.7 eV, respectively. For the C1 → C2 → C3 → C4 reaction path (Figure 8), the highest dissociation energy barrier was 1.59 eV, between those of TS5 and TS2 (TS2: 0.31 and TS5: 1.69 eV, respectively). The total energy of structure C4 was reduced by 1.34 eV, which was much higher than that of the two former systems. It showed that with the help of Cu-Pt dimer atoms, the dissociation process of CO_2_ on the A-TiO_2_(101) surface also required a relatively high-energy barrier.

We studied the adsorption and reduction performance of CO_2_ molecules on three configurations of bimetallic dimer-embedded TiO_2_(101): Zn-Cu, Zn-Pt, and Cu-Pt. We found unpaired electrons induced by the transfer of the bimetallic dimer to the O atom of the TiO_2_ substrate. While a CO_2_ molecule appeared on the TiO_2_(101) surface, a partial excess charge of the bimetallic dimer was used to absorb the CO_2_ gas molecule. However, the adsorption energy was relatively low, indicating only weaker physical adsorption. The final product was mainly CO, with the O atom bonded with the surrounding metal atom due to the CO_2_-dissociation process on the bimetallic dimer-embedded TiO_2_(101) surface.

For the CO_2_ reduction process on the Zn-Cu-embedded TiO_2_(101) surface, the highest-energy barrier was as low as 0.31 eV with the breaking of the C-O_(2)_ bond and the formation of Cu-O_(2)_, Zn-O_(2)_, and Ti-O_(2)_ bonds, which was relatively suitable to produce CO and O adatoms easily. However, relatively higher-energy barriers existed for Zn-Pt and Cu-Pt dimer-embedded TiO_2_(101) systems, namely 1.69 eV and 1.58 eV, respectively. As shown in Figure 7b and Figure 8b, the C-O_(2)_ bonds of these two systems were also broken, besides the formation of three metal-O_(2)_ bonds. However, the Zn-Pt-TiO_2_ system had higher reaction energy of 0.62 eV, promoting thermal-catalytic performance. With lower barrier energy of 0.31 eV, the Zn-Cu dimer-embedded TiO_2_(101) surface might be a much more convenient design for the CO_2_RR of the bimetallic dimer-embedded TiO_2_ substrate, as well as a more economical method compared with noble metals such as Pt, Au, and Ag. Therefore, the findings of this study might theoretically provide an effective strategy for the designing of effective CO_2_-reduction TiO_2_-based catalysts.

## 4. Conclusions

In this study, the enhancing effects of the Zn-Cu, Zn-Pt, and Cu-Pt dimers on the CO_2_-reduction reaction of the A-TiO_2_(101) surface were discussed. When the bimetallic dimer was embedded on the TiO_2_(101) surface, the configuration was relatively more stable with higher binding energy. Meanwhile, the adsorption energy of CO_2_ molecules on the bimetallic dimer-embedded TiO_2_(101) surface was relatively lower, indicating a weaker interaction. Moreover, the Bader charge analysis exhibited only 0.03 e transferring from the CO_2_ molecule to the bimetal-embedded TiO_2_(101) surface, demonstrating only physical adsorption behavior. After a lower-energy barrier, the higher adsorption energy was up to −0.46 eV for the Zn-Pt-TiO_2_ system and 0.82 eV of the Zn-Cu-TiO_2_ system, respectively. Furthermore, the surface-electronic states also implied stronger interaction between CO_2_ molecules and the bimetallic dimer-embedded TiO_2_(101) surface.

Regarding the CO_2_ dissociation process, the results showed that the dissociation barriers of the CO_2_ molecule with Zn-Cu, Zn-Pt, and Cu-Pt dimers embedded on the TiO_2_(101) surface had the values of 0.31 eV, 1.69 eV, and 1.58 eV, respectively, which benefited the CO_2_-reduction reaction (CO_2_RR) activity. Meanwhile, it was observed that the products *CO and *O* of CO_2_ reduction were firmly adsorbed on the dimer-embedded TiO_2_(101) surface. In addition, the bonds of metal-O_(2)_-metal were less than 2.00 Å, and metal-C and C-O_(1)_ bonds were relatively shortened to be 1.76 Å and 1.15 Å, respectively. From the viewpoint of the activation barrier and reaction energy, the Zn-Cu dimer-embedded TiO_2_(101) substrate was more favorable for CO_2_ direct reduction than the other two kinds of bimetallic dimer-embedded TiO_2_(101) surfaces. Meanwhile, compared with partial noble atoms such as Pt, Au, and Ag, the combination of Zn and Cu atoms was a much more suitable choice, as well as being advantageous in terms of economy with the energy barrier of 0.31 eV. Therefore, our study theoretically provided a new design of the high CO_2_RR performance of Zn-Cu dimer-embedded A-TiO_2_ substrate catalysts.

## Figures and Tables

**Figure 1 materials-15-02538-f001:**
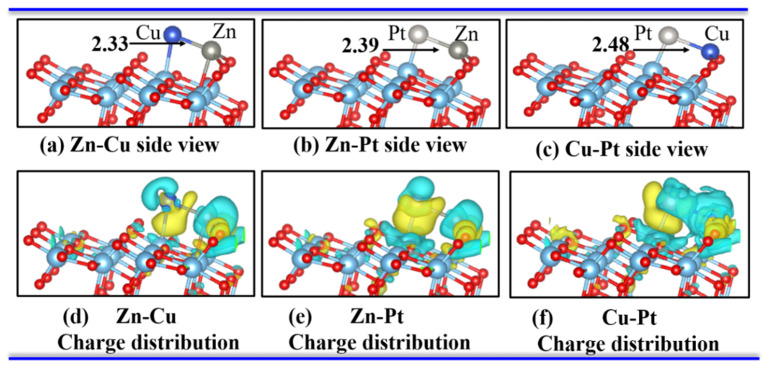
(**a**–**c**) Bimetallic-dimer interstitial configurations. (**d**–**f**) Charge density induced by Zn-Cu, Zn-Pt, and Cu-Pt interstitials of the TiO_2_(101) surface from a side view, respectively. The isosurface value was set as 0.0025 e/bohr^3^. The electron accumulation is denoted with the yellow regions; green indicates electron depletion. The Cu atom in dark blue color, Pt atom is pale, and Zn atom in light gray color.

**Figure 2 materials-15-02538-f002:**
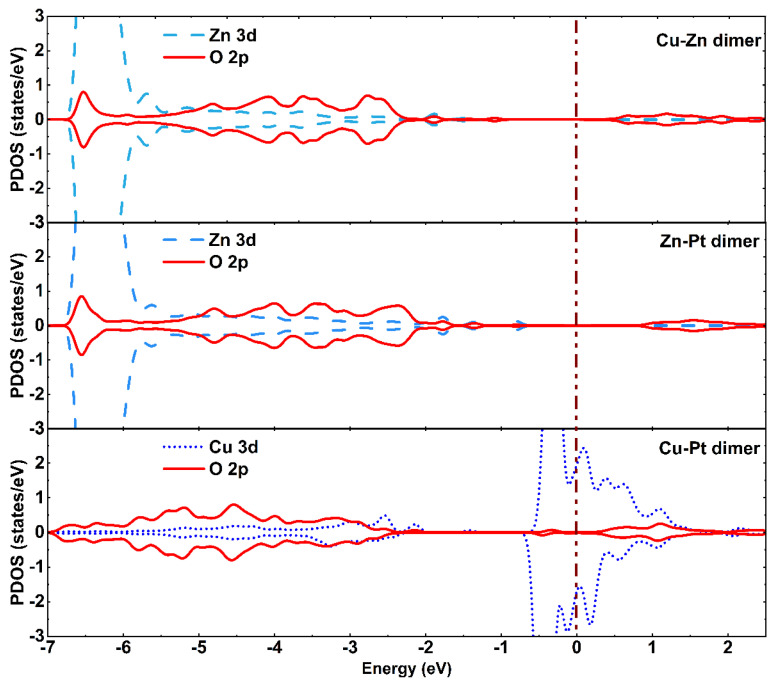
Partial density of states for the bimetal-TiO_2_ system (bimetallic dimers are Zn-Cu, Zn-Pt, and Cu-Pt). The Fermi level was set to zero and is marked with a grown dashed line.

**Figure 3 materials-15-02538-f003:**
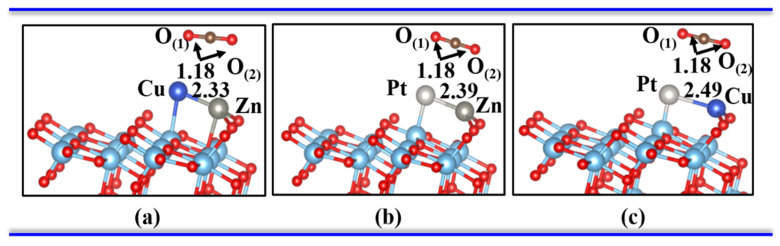
One CO_2_ molecule absorbed on the surface of the bimetal-TiO_2_(101) system from a side view. (**a**–**c**) are Cu-Zn, Pt-Zn and Pt-Cu, respectively.

**Figure 4 materials-15-02538-f004:**
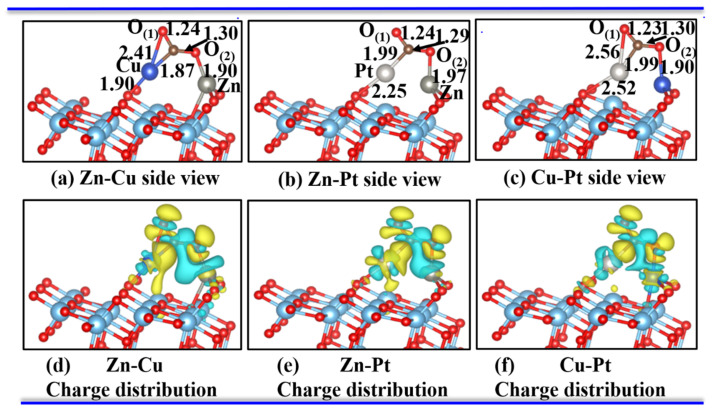
Most stable configurations (**a**–**c**) and the corresponding charge-density differences (**d**–**f**) of a CO_2_ molecule absorbed on the Zn-Cu, Zn-Pt, and Cu-Pt dimer-embedded A-TiO_2_(101) surface from a side view. The isosurface value was set as 0.0025 e/bohr^3^. The electron accumulation is denoted with the yellow regions, while the electron depletion is denoted with green regions. The Cu atom in dark blue color, Pt atom is pale, and Zn atom in light gray.

**Figure 5 materials-15-02538-f005:**
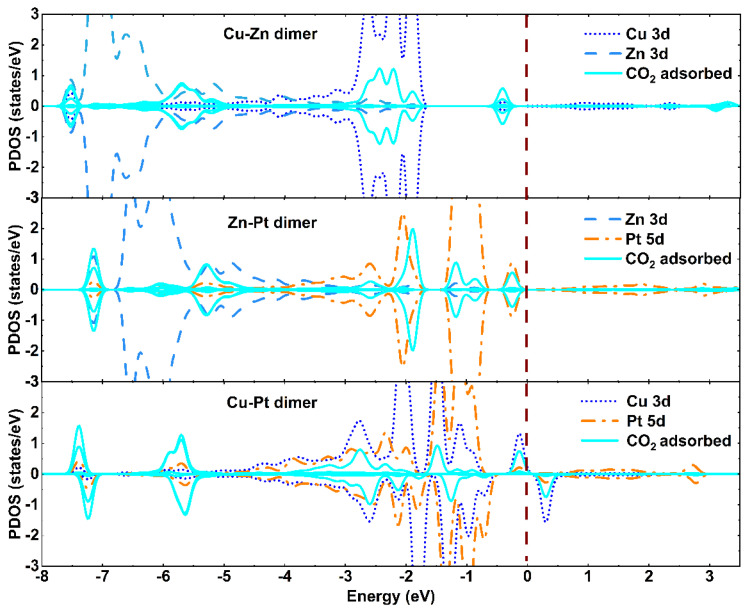
Partial density of states of the absorbed CO_2_ on bimetal-embedded TiO_2_ system (bimetallic dimers were Zn-Cu, Zn-Pt, and Cu-Pt). The Fermi level was set to zero and is marked with a grown dashed line.

**Figure 6 materials-15-02538-f006:**
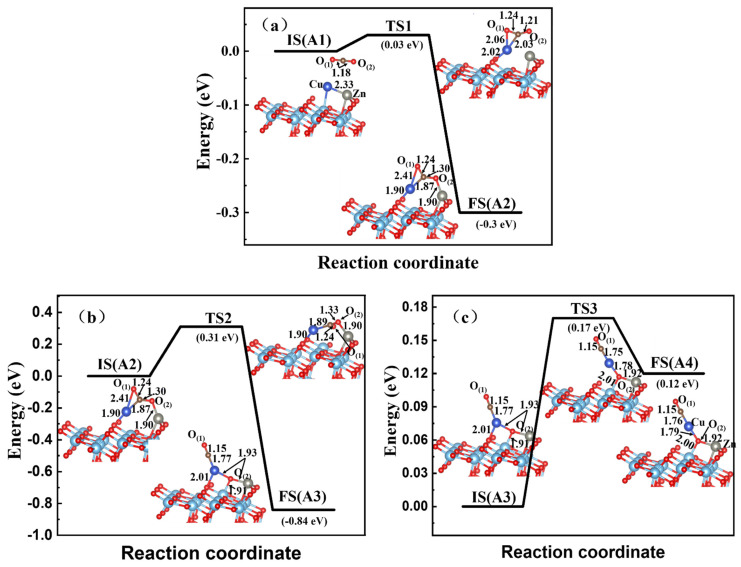
CO_2_ reduction on the Zn-Cu-embedded A-TiO_2_(101) surface. For the A-TiO_2_ system, the bond unit was Å. The Cu atom in dark blue, and Zn atom in light gray. (**a**) is the adsorption process, (**b**,**c**) are the two steps of reduction process.

**Figure 7 materials-15-02538-f007:**
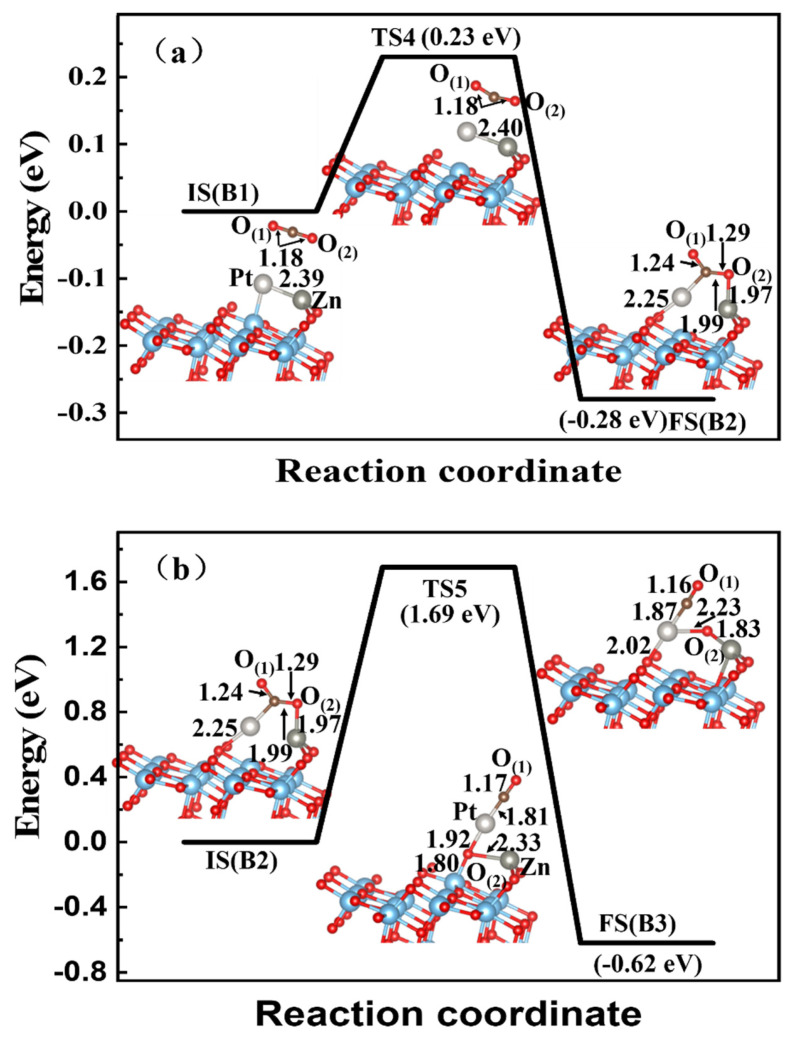
CO_2_ reduction on Zn-Pt-embedded A-TiO_2_(101) surface. The Cu atom in dark blue, Pt atom is pale, and Zn atom in light gray. (**a**) is the adsorption process, and (**b**) is the reduction process.

**Figure 8 materials-15-02538-f008:**
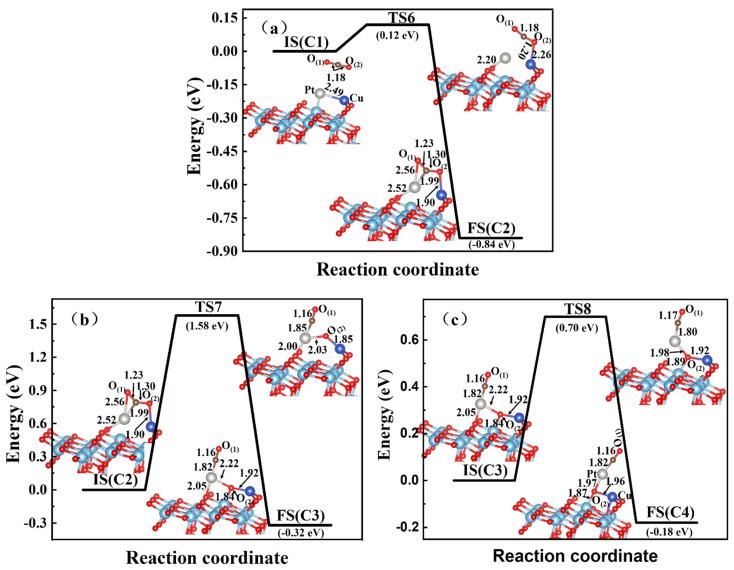
CO_2_ reduction on the Cu-Pt-embedded A-TiO_2_(101) surface. The Cu atom in dark blue, Pt atom is pale, and Zn atom in light gray. (**a**) is the adsorption process, (**b**) and (**c**) are the two steps of reduction process.

**Table 1 materials-15-02538-t001:** Calculated results for bimetal-embedded A-TiO_2_ systems, including the distance d _M-N_ between two metal atoms; d _M-O_ and d _M-N_ is the distance of metal M and O_(2)_, and Ti and metal N atom, respectively. The angle of O_2c_-M-O_2c_ is equal to ∠O_2c_-M-O_2c_. The binding energies of bimetallic atoms on A-TiO_2_(101) (E_b_) and the charge transfer from A-TiO_2_ to bimetallic atoms (∆*Q*). M stands for the former metal atom, and N stands for the later one.

Dimer	d_M-N_ (Å)	d_M-O_ (Å)	∠O_2c_-M-O_2c_ (°)	*d*_Ti-N_ (Å)	E_b_ (eV)	∆*Q* (e)
Zn-Cu	2.32	2.07	108.42°	2.62	−2.34	−0.85
Zn-Pt	2.39	2.02	114.96°	2.28	−3.60	−0.52
Cu-Pt	2.48	1.92	135.90°	2.29	−2.90	−0.60

**Table 2 materials-15-02538-t002:** Calculated results for a CO_2_ molecule on an A-TiO_2_(101) surface during the adsorption process. d_M-N_ and d_O(1)-C_ are the bond lengths between metal M and metal N, and between the O_(1)_ atom and the C atom, respectively, for chemical adsorption structure. E_CO2ads_ stands for the adsorption energy of CO_2_ adsorbed on TiO_2_(101) surface. ∆*Q* is the charge transfer from bimetallic dimer-embedded TiO_2_ to CO_2_ molecule. M stands for the former metal atom, and N stands for the latter one.

	$d_{M-N}$ (Å)	$d_{M-O_{\left(2\right)}}$ (Å)	$d_{O_{\left(1\right)}-C}$ (Å)	$d_{C-O_{\left(2\right)}}$ (Å)	∠O_2c_-M-O_2c_ (°)	$d_{C-N}$ (Å)	$d_{O_{2c}-N}$ (Å)	$E_{CO_{2}\mathrm{ads}}$ (eV)	∆Q (e)
Pure TiO_2_	-	-	1.18	1.18	180	-	-	−0.31	-
Zn-Cu	3.45	1.90	1.24	1.30	125.92	1.87	1.90	−0.36	0.82
Zn-Pt	2.68	1.97	1.24	1.29	130.90	1.99	2.25	−0.46	0.67
Cu-Pt	2.63	1.90	1.23	1.30	132.32	1.99	2.52	−0.97	0.54

## Data Availability

The data presented in this study are available within the article.

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
