# Peer review of "Effect of Bimetallic Dimer-Embedded TiO2(101) Surface on CO2 Reduction: The First-Principles Calculation"

_materials, 2022, doi:10.3390/ma15072538_

Round 1

Reviewer 1 Report

Reviewer report for Chongyang Li et al. “Effect of Bimetallic Dimer-Embedded TiO2 (101) Surface on CO2 Reduction: The First-Principles Calculation”

The work gives a theoretical perspective on the reduction of CO2 on TiO2(101) surfaces with deposited Zn-Cu, Zn-Pt, and Zn-Pd dimers. The computational work is done very well, but the presentations lacks a bit.

I suggest a minor revision prior publication.

Please take care of super script indeces for CO2 and TiO2, e.g., lines 131-139.

Figure 2 and 5: please, introduce also different line types for the differently colored PDOS lines.

Figures 1, 3, 4, 6, 7, 8: These figures are too small, and a lot of the data need to be put into tables, e.g. C-O bond lengths, O-C-O angles, distances of the extra metals, change of geometry TiO2-dimer-[O,C,O], etc. This will help the presentation of your results.

Author Response

Thank you very much for your constructive comments and suggestions on our manuscript entitled “Effect of Bimetallic Dimer-Embedded TiO2(101) Surface on CO2 Reduction: The First-Principles Calculation” (manuscript number: materials-1629412.) These comments are very helpful to revise our paper and improve the quality of our paper. We have tried our best to make corrections which we hope meet the approval for publication. The main corrections in the paper and the responses to the reviewers’ comments are listed as follows:

Report #1:

Reviewer report for Chongyang Li et al. “Effect of Bimetallic Dimer-Embedded TiO2 (101) Surface on CO2 Reduction: The First-Principles Calculation”

The work gives a theoretical perspective on the reduction of CO2 on TiO2(101) surfaces with deposited Zn-Cu, Zn-Pt, and Zn-Pd dimers. The computational work is done very well, but the presentations lacks a bit.

I suggest a minor revision prior publication.

  1. Please take care of super script indeces for CO2 and TiO2, e.g., lines 131-139.

Our response: Thank you very much for your suggestions. We have made the corresponding revisions. Please see lines 129-137.

  1. Figure 2 and 5: please, introduce also different line types for the differently colored PDOS lines.

Our response: Thank you very much for your suggestions. We have made the corresponding revisions, such as the dash lines stand for the 3d orbit of Zn, short dot lines stand for the 3d orbit of Cu and dash dot lines for the 5d orbit of Pt. Please see Figure 2 and 5.

  1. Figures 1, 3, 4, 6, 7, 8: These figures are too small, and a lot of the data need to be put into tables, e.g. C-O bond lengths, O-C-O angles, distances of the extra metals, change of geometry TiO2-dimer-[O,C,O], etc. This will help the presentation of your results.

Our response: Thank you very much for your suggestions. We have added C-O bond lengths, O-C-O angles, dM-O(2), dC-O(2), ∠O2c-M-O2c, dC-N and dO2c-N. At the same, we have enlarged these six figures to better exhibit the results. Please Table 2, Figures 1, 3, 4, 6, 7 and 8.

We would like to take this opportunity to express again our sincere appreciation to the reviewer’s comments on our paper. These comments are very important for further improving the quality of our paper.

Sincerely yours,

Chongyang Li, Bin Zhao, Zhiquan Chen

Reviewer 2 Report

The manuscript submitted by Chen et al. entitled 'Effect of Bimetallic Dimer-Embedded TiO2(101) Surface on CO2 Reduction: The First-Principles Calculation' (Manuscript ID: materials-1629412) towards the publication in Materials describes the a new and first research work on the calculation to explore the effectiveness in CO2 reduction on bimetal dimer (Zn-Cu, Zn-Pt and Cu-Pt)-embedded anatase TiO2 (A-TiO2)(101) surface. The calculations and their discussions are found to be logical and this work showed the possibility to achieve a highly economical bimetalic (such as Zn-Cu) dimer embedded A-TiO2 based catalyst for the CO2 reduction reaction (CO2RR). Further, the current state of the earth needs to develop such cost effective processes for the utilization of CO2 there by decreasing the global warming. By these observations this article is recommended for the publication of this article in Materials. The manuscript was prepared very well, however the following are some general comments to the authors which are to be considered while revising the manuscript. 

  1. Author name Zhiquan,* Chen5,* may be corrected.
  2. The duplication of the addresses of the authors must be avoided.
  3. The introduction part of the article should be included some more discussion about the rise of global population and related consequences including greenhouse effect and in this regard the articles: i. Desa, UN, World population prospects 2019: Highlights. New York, NY: United Nations Department for Economic and Social Affairs, 2019; ii. Millati, R.; Cahyono, R.B.; Ariyanto, T.; Azzahrani, I.N.; Putri, R.U.; Taherzadeh, M.J.; Sustainable resource recovery and zero waste approaches. Elsevier, New York, 2019; iii. Sustain. Chem. Pharm., 2021, 19, 100371; iv. Environ. Chem. Lett., 2021, 19, 3887–3950; v. Bioresour. Technol. 2020, 317, 123987; vi. Green Chem. Lett. Rev., 2021, 14, 710–722; vii. Mol. Catal., 2021, 511, 111719; and viii. Egypt. J. Petrol 2018, 27, 1275–1290; ix. Sustain. Chem. Pharm., 2022, 25, 100610 should be adopted and are cited to improve the reference section of the manuscript.
  4. Page no.s 3 and 4: ‘CO2’, ‘TiO2’, ‘M-O2c’ and ‘O2c-M-O2c’ should be corrected to ‘CO2’, ‘TiO2’, ‘M-O2c’ and ‘O2c-M-O2c’.
  5. Authors names and initials of all the references should be adopted the Materials journal format.
  6. All the names of the journals in reference section should be corrected as per the present journal’s convention.
  7. Reference 18, 21: Angewandte Chemie International Edition reference should also be provided.

Author Response

Thank you very much for your constructive comments and suggestions on our manuscript entitled “Effect of Bimetallic Dimer-Embedded TiO2(101) Surface on CO2 Reduction: The First-Principles Calculation” (manuscript number: materials-1629412.) These comments are very helpful to revise our paper and improve the quality of our paper. We have tried our best to make corrections which we hope meet the approval for publication. The main corrections in the paper and the responses to the reviewers’ comments are listed as follows:

Report #2:

The manuscript submitted by Chen et al. entitled 'Effect of Bimetallic Dimer-Embedded TiO2(101) Surface on CO2 Reduction: The First-Principles Calculation' (Manuscript ID: materials-1629412) towards the publication in Materials describes the a new and first research work on the calculation to explore the effectiveness in CO2 reduction on bimetal dimer (Zn-Cu, Zn-Pt and Cu-Pt)-embedded anatase TiO2 (A-TiO2)(101) surface. The calculations and their discussions are found to be logical and this work showed the possibility to achieve a highly economical bimetalic (such as Zn-Cu) dimer embedded A-TiO2 based catalyst for the CO2 reduction reaction (CO2RR). Further, the current state of the earth needs to develop such cost effective processes for the utilization of CO2 there by decreasing the global warming. By these observations this article is recommended for the publication of this article in Materials. The manuscript was prepared very well, however the following are some general comments to the authors which are to be considered while revising the manuscript.

  1. Author name Zhiquan,* Chen5,* may be corrected.

Our response: Thank you very much for your suggestions. We have revised as Zhiquan Chen5,*. Please see line 4.

  1. The duplication of the addresses of the authors must be avoided.

Our response: Thank you very much for your suggestions. We have made the corresponding revisions. Please see lines 5-15

  1. The introduction part of the article should be included some more discussion about the rise of global population and related consequences including greenhouse effect and in this regard the articles: i. Desa, UN, World population prospects 2019: Highlights. New York, NY: United Nations Department for Economic and Social Affairs, 2019; ii. Millati, R.; Cahyono, R.B.; Ariyanto, T.; Azzahrani, I.N.; Putri, R.U.; Taherzadeh, M.J.; Sustainable resource recovery and zero waste approaches. Elsevier, New York, 2019; iii. Sustain. Chem. Pharm., 2021, 19, 100371; iv. Environ. Chem. Lett., 2021, 19, 3887–3950; v. Bioresour. Technol. 2020, 317, 123987; vi. Green Chem. Lett. Rev., 2021, 14, 700–712; vii. Mol. Catal., 2021, 511, 111719; and viii. Egypt. J. Petrol 2018, 27, 1275–1290; ix. Sustain. Chem. Pharm., 2022, 25, 100610 should be adopted and are cited to improve the reference section of the manuscript.

Our response: Thank you very much for your suggestions. We have revised and added these papers. Please see the introduction part.

  1. Page no.s 3 and 4: ‘CO2’, ‘TiO2’, ‘M-O2c’ and ‘O2c-M-O2c’ should be corrected to ‘CO2’, ‘TiO2’, ‘M-O2c’ and ‘O2c-M-O2c’.

Our response: Thank you very much for your suggestions. We have made the corresponding revisions. Please see lines 129-137 of page no.s 3 and 4.

  1. Authors names and initials of all the references should be adopted the Materials journal format.

Our response: Thank you very much for your suggestions. We have made the corresponding revisions. Please see all the references.

  1. All the names of the journals in reference section should be corrected as per the present journal’s convention.

Our response: Thank you very much for your suggestions. We have made the corresponding revisions. Please see all the references.

  1. Reference 18, 21: Angewandte Chemie International Edition reference should also be provided.

Our response: Thank you very much for your corrections. We have made the corresponding revisions. Please see reference 27 and 30.

We would like to take this opportunity to express again our sincere appreciation to the reviewer’s comments on our paper. These comments are very important for further improving the quality of our paper.

Sincerely yours,

Chongyang Li, Bin Zhao, Zhiquan Chen

Reviewer 3 Report

Ms. Ref. No.: materials-1629412

The manuscript entitled "Effect of Bimetallic Dimer-Embedded TiO2 (101) Surface on CO2 Reduction: The First-Principles Calculation " was studied.

The DFT calculations were performed correctly. Obtained results are valuable and well. However, some comments should be addressed in revised manuscript to fulfil the requirement of a proper manuscript.

It needs a minor revision regarding following comments:

  • It was helpful if the DFT calculation for CO2 adsorption on pure TiO2 was added to table 1 and 2.
  • Page 5, Line 7, “3d orbital electrons of Zn, Zn, and Cu atoms,” one “Zn” should be omitted.
  • Page 6, Paragraph 2, Line 4, “were absorbed” should be corrected as “were adsorbed”.

Author Response

Thank you very much for your constructive comments and suggestions on our manuscript entitled “Effect of Bimetallic Dimer-Embedded TiO2(101) Surface on CO2 Reduction: The First-Principles Calculation” (manuscript number: materials-1629412.) These comments are very helpful to revise our paper and improve the quality of our paper. We have tried our best to make corrections which we hope meet the approval for publication. The main corrections in the paper and the responses to the reviewers’ comments are listed as follows:

Report#3:

Ms. Ref. No.: materials-1629412

The manuscript entitled "Effect of Bimetallic Dimer-Embedded TiO2 (101) Surface on CO2 Reduction: The First-Principles Calculation " was studied.

The DFT calculations were performed correctly. Obtained results are valuable and well. However, some comments should be addressed in revised manuscript to fulfil the requirement of a proper manuscript.

It needs a minor revision regarding following comments:

  1. It was helpful if the DFT calculation for CO2 adsorption on pure TiO2 was added to Table 1 and 2.

Our response: Thank you very much for your suggestion. Table 1 is only listed for the structure parameters of bimetallic dimer embedded TiO2, and Table 2 is the data of CO2 adsorption on A-TiO2(101) surface. We have calculated this case of CO2 adsorbed on pure TiO2, and the corresponding data was added to table 2. Please see Table 2.

  1. Page 5, Line 7, “3d orbital electrons of Zn, Zn, and Cu atoms,” one “Zn” should be omitted.

Our response: Thank you very much for your corrections. We have made the corresponding revisions. Please see Page 5, Line 8.

  1. Page 6, Paragraph 2, Line 4, “were absorbed” should be corrected as “were adsorbed”.

Our response: Thank you very much for your corrections. We have made the corresponding revisions. Please see Page 6, Paragraph 2, Line 4.

We would like to take this opportunity to express again our sincere appreciation to the reviewer’s comments on our paper. These comments are very important for further improving the quality of our paper.

Sincerely yours,

Chongyang Li, Bin Zhao, Zhiquan Chen

Reviewer 4 Report

The manuscript reports first-principles calculation used to explore the effect of a novel heterogeneous catalyst  for CO2 reduction. The main question addressed by this research is the report that a non-noble Zn-Cu alloy could be a more suitable and economical choice when combined with TiO2 catalysts. The findings are unprecedented, and the topic original or relevant to the field. CO2 reduction reaction still suffers from low faradaic efficiency and low conversion efficiency. Transition metal (TM)-based catalysts have been employed to  improve the performance, but the use of Zn-Cu cocatalyst is unprecedented. The conclusions are consistent with the evidence presented and present a clear advances over the state of the art in the field. The paper is well written and presented, with appropriate and well designed figures and tables, the results are relevant for all involved in the described type of catalysis and the paper can have a broad impact on CO2 scientific community. I therefore recommend acceptance, provided the authors add relevant citations about surface chemistry: doi: 10.1021/acs.jpcc.5b05547; doi: 10.1021/nn4035684; 10.1021/cm010431w

Author Response

Thank you very much for your constructive comments and suggestions on our manuscript entitled “Effect of Bimetallic Dimer-Embedded TiO2(101) Surface on CO2 Reduction: The First-Principles Calculation” (manuscript number: materials-1629412.) These comments are very helpful to revise our paper and improve the quality of our paper. We have tried our best to make corrections which we hope meet the approval for publication. The main corrections in the paper and the responses to the reviewers’ comments are listed as follows:

Report#4:

The manuscript reports first-principles calculation used to explore the effect of a novel heterogeneous catalyst for CO2 reduction. The main question addressed by this research is the report that a non-noble Zn-Cu alloy could be a more suitable and economical choice when combined with TiO2 catalysts. The findings are unprecedented, and the topic original or relevant to the field. CO2 reduction reaction still suffers from low faradaic efficiency and low conversion efficiency. Transition metal (TM)-based catalysts have been employed to improve the performance, but the use of Zn-Cu cocatalyst is unprecedented. The conclusions are consistent with the evidence presented and present a clear advances over the state of the art in the field. The paper is well written and presented, with appropriate and well designed figures and tables, the results are relevant for all involved in the described type of catalysis and the paper can have a broad impact on CO2 scientific community. I therefore recommend acceptance, provided the authors add relevant citations about surface chemistry: doi: 10.1021/acs.jpcc.5b05547; doi: 10.1021/nn4035684; 10.1021/cm010431w

Our response: Our response: Thank you very much for your affirmation and suggestions. We have revised and added these papers. Please see the introduction part.

We would like to take this opportunity to express again our sincere appreciation to the reviewer’s comments on our paper. These comments are very important for further improving the quality of our paper.

Sincerely yours,

Chongyang Li, Bin Zhao, Zhiquan Chen

Reviewer 5 Report

This manuscript deals with modelling the effect of a bimetallic dimer-embedded (Zn-Cu, Zn-Pt and Cu-Pt) anatase TiO2(101) surface on reduction of carbon dioxide by using quantum chemical calculations (a DFT approach) focusing on the computing adsorption energies, activation barriers and reaction energies. Since the theoretic calculcations could provide a new design and novel improved materials for developing high performance reduction of CO2, this investigation deserves attention.

The manuscript is fairly well written and the results are clearly presented. However, some minor revisions should be made before publication:

1) p. 1 line 82 The authors write that “The energy barrier of the Zn-Cu dimer−embedded TiO2 (101) surface was as low as 0.31 eV, which was a much more suitable choice as well as advantageous in terms of economy.”, but this contains some new results which should be moved and shown in the Results and Discussion part.

2) ps. 3–11 The sizes of Figs. 1, 3, 4 and 6–8 should be increased, because they are hardly readable in these forms.

3) ps. 13–14 Style of the references does not meet the requirements of journal Materials. For example, “… Journal of Cleaner Production, 2018, 200, 791-808. or Nature communications, 2014, 5, 4948.” instead of J. Cleaner Prod., 2018, 200, 791–808., Nat. Commun., 2014, 5, 4948. Please, check all of them and modify them accordingly.

4) The English also needs improvements. There are some, typical grammar or typing mistakes:

p. 1 line 21 and elsewhere „… bimetal …” instead of  bimetallic

       line 22 and elsewhere „… TiO2 (101) ...”  instead of  TiO2(101)

       line 40 and elsewhere „…  contributor[2].”  instead of  contributor [2].

       line 47 „… employed …”  instead of  applied

p. 3 line 125 and elsewhere „… Side view …” instead of side view

       line 131 and elsewhere „… CO2 …, … TiO2 (101) …”  instead of  CO2, TiO2(101)

p. 4 line 138 and elsewhere „… M-O2c, O2c-M-O2c ...” instead of M-O2c, O2c-M-O2c

p. 5 line 156 „… Further, The Partial density ...” instead of Furthermore, the partial density

p. 6 line 210 „Besides the calculations of charge distribution, ...” instead of In addition to the calculations of charge distribution,

p. 10 line 296 „Similar to ...” instead of Similarly to

Author Response

Thank you very much for your constructive comments and suggestions on our manuscript entitled “Effect of Bimetallic Dimer-Embedded TiO2(101) Surface on CO2 Reduction: The First-Principles Calculation” (manuscript number: materials-1629412.) These comments are very helpful to revise our paper and improve the quality of our paper. We have tried our best to make corrections which we hope meet the approval for publication. The main corrections in the paper and the responses to the reviewers’ comments are listed as follows:

Report#5:

This manuscript deals with modelling the effect of a bimetallic dimer-embedded (Zn-Cu, Zn-Pt and Cu-Pt) anatase TiO2(101) surface on reduction of carbon dioxide by using quantum chemical calculations (a DFT approach) focusing on the computing adsorption energies, activation barriers and reaction energies. Since the theoretic calculcations could provide a new design and novel improved materials for developing high performance reduction of CO2, this investigation deserves attention.

The manuscript is fairly well written and the results are clearly presented. However, some minor revisions should be made before publication:

  1. p. 1 line 82 The authors write that “The energy barrier of the Zn-Cu dimer−embedded TiO2 (101) surface was as low as 0.31 eV, which was a much more suitable choice as well as advantageous in terms of economy.”, but this contains some new results which should be moved and shown in the Results and Discussion part.

Our response: Thank you very much for your corrections. We have made the corresponding revisions. Please see lines 347-352 and 375-376.

  1. ps. 3–11 The sizes of Figs. 1, 3, 4 and 6–8 should be increased, because they are hardly readable in these forms.

Our response: Thank you very much for your suggestions. We have enlarged and replotted the Figs. 1, 3, 4 and 6-8. Please see Figs. 1, 3, 4 and 6-8

  1. ps. 13–14 Style of the references does not meet the requirements of journal Materials. For example, “… Journal of Cleaner Production, 2018, 200, 791-808. or Nature communications, 2014, 5, 4948.” instead of J. Cleaner Prod., 2018, 200, 791–808., Nat. Commun., 2014, 5, 4948. Please, check all of them and modify them accordingly.

Our response: Thank you very much for your suggestions. We have made the corresponding revisions. Please see all the references.

  1. The English also needs improvements. There are some, typical grammar or typing mistakes:

  1. 1 line 21 and elsewhere „… bimetal …” instead of bimetallic

Our response: Thank you very much for your corrections. We have made the corresponding revisions throughout the text.

       line 22 and elsewhere „… TiO2 (101) ...”  instead of  TiO2(101)

Our response: Thank you very much for your corrections. We have made the corresponding revisions throughout the text.

       line 40 and elsewhere „…  contributor[2].”  instead of  contributor [2].

Our response: Thank you very much for your corrections. We have made the corresponding revisions throughout the text.

       line 47 „… employed …”  instead of  applied

Our response: Thank you very much for your corrections. We have made the corresponding. Please see line 44.

  1. 3 line 125 and elsewhere „… Side view …” instead of side view

Our response: Thank you very much for your corrections. We have made the corresponding revisions throughout the text.

       line 131 and elsewhere „… CO2 …, … TiO2 (101) …”  instead of  CO2, TiO2(101)

Our response: Thank you very much for your corrections. We have made the corresponding. Please see lines 129-137.

  1. 4 line 138 and elsewhere „… M-O2c, O2c-M-O2c ...” instead of M-O2c, O2c-M-O2c

Our response: Thank you very much for your corrections. We have made the corresponding. Please see lines 129-137.

  1. 5 line 156 „… Further, The Partial density ...” instead of Furthermore, the partial density

Our response: Thank you very much for your corrections. We have made the corresponding. Please see line 154.

  1. 6 line 210 „Besides the calculations of charge distribution, ...” instead of In addition to the calculations of charge distribution,

Our response: Thank you very much for your corrections. We have made the corresponding. Please see line 208.

  1. 10 line 296 „Similar to ...” instead of Similarly to

Our response: Thank you very much for your corrections. We have made the corresponding. Please see line 293.

We would like to take this opportunity to express again our sincere appreciation to the reviewer’s comments on our paper. These comments are very important for further improving the quality of our paper.

Sincerely yours,

Chongyang Li, Bin Zhao, Zhiquan Chen
